# An Overview of the Immune Modulatory Properties of Long Non-Coding RNAs and Their Potential Use as Therapeutic Targets in Cancer

**DOI:** 10.3390/ncrna9060070

**Published:** 2023-11-11

**Authors:** Moises Martinez-Castillo, Abdelrahman M. Elsayed, Gabriel López-Berestein, Paola Amero, Cristian Rodríguez-Aguayo

**Affiliations:** 1Department of Experimental Therapeutics, The University of Texas MD Anderson Cancer Center, Houston, TX 77054, USA; mpcastillo@mdanderson.org (M.M.-C.); glopez@mdanderson.org (G.L.-B.); pamero@mdanderson.org (P.A.); 2Liver, Pancreas and Motility Laboratory, Unit of Research in Experimental Medicine, School of Medicine, Universidad Nacional Autónoma de México (UNAM), Mexico City 06726, Mexico; 3Department of Pharmacology & Toxicology, Faculty of Pharmacy, Al-Azhar University, Cairo 11754, Egypt; abdopharmacy@azhar.edu.eg; 4Havener Eye Institute, Department of Ophthalmology and Visual Science, The Ohio State University Wexner Medical Center, Columbus, OH 43210, USA; 5Center for RNA Interference and Non-Coding RNA, The University of Texas MD Anderson Cancer Center, 1515 Holcombe Blvd, Houston, TX 77030, USA

**Keywords:** lncRNA, ncRNA, cancer, immune response, biomarkers, immunotherapy

## Abstract

Long non-coding RNAs (lncRNAs) play pivotal roles in regulating immune responses, immune cell differentiation, activation, and inflammatory processes. In cancer, they are gaining prominence as potential therapeutic targets due to their ability to regulate immune checkpoint molecules and immune-related factors, suggesting avenues for bolstering anti-tumor immune responses. Here, we explore the mechanistic insights into lncRNA-mediated immune modulation, highlighting their impact on immunity. Additionally, we discuss their potential to enhance cancer immunotherapy, augmenting the effectiveness of immune checkpoint inhibitors and adoptive T cell therapies. LncRNAs as therapeutic targets hold the promise of revolutionizing cancer treatments, inspiring further research in this field with substantial clinical implications.

## 1. Introduction

Transcriptome analysis of the human genome has revealed that the vast majority of RNA transcripts do not encode proteins; thus, they were named non-coding RNAs (ncRNAs) [1,2]. ncRNAs can be divided into two major categories based on nucleotide length: small ncRNAs and long ncRNAs (lncRNAs) [2,3,4]. Small ncRNAs are transcripts less than 200 nucleotides in length; some examples are transfer RNAs (tRNAs), small nuclear RNAs (snRNAs), small nucleolar RNAs (snoRNAs), small interfering RNAs (siRNAs), microRNAs (miRNAs), and piwi-interacting RNAs (piRNAs) [4]. Among those, siRNAs and miRNAs play an essential role in tumorigenesis by regulating the expression of tumor suppressor genes or oncogenes. Furthermore, several miRNAs can be employed as prognostic and/or diagnostic biomarkers in cancer [5]. From a mechanistic perspective, siRNAs and miRNAs complementarily bind to target mRNAs, resulting in the degradation of target mRNAs and therefore inducing gene knockdown [4].

Long ncRNAs (lncRNAs) are transcripts that cannot encode proteins, which have been classified, according to genomic localization and orientation, into intergenic lncRNAs (lincRNAs), intronic lncRNAs, natural antisense transcripts, pseudogenes (without producing protein), and retrotransposons [6]. Interestingly, lncRNAs can regulate the expression of nearby genes on the same allele (in cis) or regulate genes at other genomic locations across the genome (in trans) [7]. In general, lncRNAs can regulate various biological processes such as proliferation, differentiation, and cell development [2]. Regarding localization, lncRNAs have been found to localize at both nuclear and cytoplasmic compartments where they bind to DNA, RNA, or proteins to exert their functions. In the nucleus, they interact with transcription factors, the chromatin-modifying complex, or ribonucleoproteins, thereby modulating the transcription of target genes [8,9]. Meanwhile, in the cytoplasm, lncRNAs regulate the stability and function of various mRNAs and proteins [10,11].

In the context of biological functions, lncRNAs have been classified into four major archetypes of molecular mechanisms: signals, decoys, guides, and scaffolds [12,13] (Figure 1).

Signals: Some lncRNAs function as signals by serving as indicators or markers for specific cellular processes. Given that lncRNA expression is sensitive to certain external stimuli, some lncRNAs can be used as signals for certain diseases. They might respond to certain stimuli or environmental changes, triggering downstream cellular events. These lncRNAs often play a role in regulatory networks, alerting the cell to adapt to various conditions. For instance, lncRNA-p21 acts as a transcriptional repressor of p53, a major tumor suppressor protein, and therefore lnc-p21 can be used as a signal for apoptosis resistance and defective repair of damaged DNA [14].

Guides: Some lncRNAs function as guide molecules by binding to enzymes or regulatory proteins. Guide lncRNAs provide targeting information and guide other molecules to specific genomic loci either in adjacent (cis) or distant (trans) locations from their point of transcription. They can direct chromatin-modifying complexes, transcription factors, co-activators, and co-repressors and direct them to their target sites, where they can regulate gene expression or other regulatory factors to their appropriate binding sites on DNA, thereby influencing gene expression patterns and cellular outcomes. [12,15]. An example of a guide lncRNA is *HOTAIR*, an extensively studied lncRNA. *HOTAIR* steers the chromatin modifier polycomb repressive complex 2 (PRC2) in trans to the developmental HOXD locus and, when excessively expressed towards cancer-associated genes, causes gene repression [16]. Another example is the lncRNA *MEG3*, which employs triple-helix interactions with underlying DNA to recruit *PRC2* to targeted genes [17]. An intriguing instance of three-dimensional organization-mediated chromosomal targeting involves *Firre*. Derived from a genomic site that evades inactivation by the X chromosome, *Firre* orchestrates nuclear domains in trans via heterogenous nuclear ribonucleoprotein hnRNP U interactions, facilitating the co-localization of diverse chromosomal sites across chromosomes 2, 9, 15, and 17 [18]. Moreover, *Firre* operates in cis, upholding X chromosome inactivation by positioning the inactive X chromosome near the nucleolus and conserving the trimethylation of histone H3 at lysine 27 (H3K27me3). *Firre* is therefore speculated to establish specific chromosomal domains within the nucleus under sequence-specific interactions, possibly serving as a signal for localizing particular subcompartments. These precise targeting mechanisms of guide lncRNAs are triggered by interactions involving RNA–DNA, RNA–RNA, and RNA–protein interactions [18].

Decoys: Some lncRNAs act as decoys. Decoy lncRNAs act as molecular sponges by sequestering regulatory molecules, such as RNA-binding proteins, transcription factors, or microRNAs. This action prevents these molecules from interacting with their intended targets, thereby influencing gene expression regulation. Decoy lncRNAs help modulate cellular processes by diverting regulatory molecules from their intended targets [12], exerting negative control over their functional counterparts. For instance, in response to DNA damage, the lncRNA *PANDA* binds with the transcription factor *NF-YA* to counteract p53-induced apoptosis. While *NF-YA* activates key apoptotic and cell senescence genes, *PANDA* binding to *NF-YA* redirects its association from gene chromatin, subsequently diminishing the expression of genes related to apoptosis and senescence [19,20,21]

Scaffolds: Scaffold lncRNAs serve as physical platforms or structures that gather multiple molecules into a complex. They provide a spatial arrangement for interactions between proteins, nucleic acids, and other molecules. These interactions can lead to the assembly of functional complexes that carry out specific cellular tasks [12,15]. For example, the telomerase RNA *TERC* is responsible for the assembly of the telomerase complex responsible for maintaining telomere ends. This complex integrates reverse transcriptase activity with telomere-targeting proteins within a single ribonucleoprotein (RNP) structure [22]. *TERC* has been a foundational model to investigate whether recently identified lncRNAs can form stable, uniform RNPs. However, limited evidence supports the notion that newly discovered lncRNAs function as stable molecular scaffolds akin to TERC. Instead, lncRNAs might engage in more dynamic, low-affinity interactions with proteins, resembling the maturation process of mRNAs. This perspective can account for the functional interactions observed between lncRNAs and mRNA biogenesis factors such as hnRNPs. This dynamic scaffold concept aligns with the diverse, substoichiometric factors identified through lncRNA isolation and proteomic analysis, such as in the case of *Xist* RNA [23,24,25,26].

The classification of lncRNAs could change in the future given the reports of novel functions of lncRNAs. In this context, novel technologies such as ribosome profiling sequencing and ribosome-nascent chain complex sequencing have allowed for the identification of short open reading frames (ORFs) within lncRNA sequences that encode for micropeptides [27]. Recently, Wang et al. identified differentially expressed long non-coding RNAs (lncRNAs) potentially harboring protein-coding capabilities within the challenging realm of triple-negative breast cancer (TNBC), renowned for its dire clinical prognosis. Employing a comprehensive approach, they employed eight distinct assays to validate the existence of a 60-amino acid peptide named ASRPS, which originates from the lncRNA *LINC00908*. Their meticulous methodology encompassed various stages, including (1) the identification of open reading frames (ORFs) through *ORFFinder*, (2) correlating ORFs with ribosome profiling data from the GWIPS-viz database, (3) creating tagged ORFs in pcDNA3.1 for Western blotting, (4) testing the functional initiation codon through GFPmut fusion, (5) profiling polysome for endogenous *ASRPS* expression, (6) detecting micropeptide and lncRNA co-expression via in situ hybridization, and (7) producing a specific rabbit polyclonal antibody for *ASRPS* identification. To dispel any doubts regarding ASRPS being a byproduct of lengthier protein processing, the authors employed antisense oligos to precisely inhibit lncRNA *ORF* translation. This extensive research added to our understanding of *ASRPS*, considerably bolstering the confidence of readers in acknowledging the significance of this newfound short peptide [28]. The specific roles of micropeptides have recently been a spotlight in research; some putative roles include regulation of cell growth, DNA repair, tissue regeneration, immune response, and cancer [29]. Thus, micropeptides need to be included in future research evaluating lncRNA functions.

Presently, various preclinical and clinical studies have shown that dysregulated expression of lncRNA is associated with the development of several diseases, including cancer. In cancer, lncRNAs can act like tumor suppressor genes [30] or oncogenes [31] depending on which downstream target pathway they regulate. Furthermore, the expression profile of certain lncRNAs can be used as biomarkers for disease progression, survival, and chemoresistance [5,32]. Further studies revealed that lncRNAs are implicated in the regulation of inflammatory signaling pathways, innate immune response, and T cell differentiation and activation [33,34,35]. In this review, we summarize the current state of knowledge regarding the pivotal role of lncRNAs in regulating immune response in cancer.

## 2. lncRNAs and Cancer

Cancer is the second leading cause of death on a global scale. In the US, the lifetime probability of developing cancer is ~44% for men and ~38% for women, respectively [36]. In addition to the significant role of lncRNAs in orchestrating multiple biological functions, lncRNAs play a significant role in carcinogenesis, and dysregulated expression of lncRNAs has been detected in a wide array of cancers. Furthermore, some lncRNAs can regulate the activity of oncogenes, whereas others can act like tumor suppressor genes [37,38,39]. In 2011, almost 200 putative lncRNAs derived from promoter regions of cell cycle genes were identified during cell cycle progression, and their expression profiles showed alterations under certain oncogenic stimuli, stem cell differentiation, or DNA damage [19]. The newly discovered lncRNAs are more recognized as active molecules instead of “transcriptional noise” and accumulating evidence indicates that some of them play critical roles in carcinogenesis by influencing tumor cell proliferation [40]. In the context of the tumor-promoting actions of lncRNAs, Elsayed et al. revealed that the lncRNA *PRKAR-1B AS2* promotes tumor growth and survival of ovarian cancer and that knockdown of *PRKAR1B-AS2* by a specific siRNA reduced tumor growth and sensitized the response to cisplatin in both in vitro and in vivo mouse models of ovarian cancer. Mechanistically, *PRKAR1B-AS2* promotes tumor growth, at least in part, by positively regulating the PI3K/AKT/mTOR pathway [39]. Certain lncRNAs have been found to regulate apoptosis, a programmed cell death process. In this setting, two lncRNAs with anti-apoptotic functions were identified in prostate and squamous carcinoma cells: *PCGEM1* (prostate-specific transcript 1) and *CUDR* (cancer upregulated drug resistant). Multiple analyses by Northern blot have supported the exclusive expression of *PCGEM* in the human prostate; additionally, the overexpression of this oncogenic lncRNA has been related to the risk of prostate cancer [41]. Moreover, a functional study of *PCGEM1* demonstrated that the overexpression of *PCGEM1* in LNCaP cells (lymph node carcinoma of the prostate) results in apoptosis inhibition induced by doxorubicin by mitigating P53 and p21^waf1/Cip1^ induction [42].

## 3. lncRNAs as Tumor Biomarkers

lncRNAs have garnered considerable attention as potential tumor biomarkers due to their distinct expression patterns in various types of cancers. In addition to the specific and sensitive expression profile of lncRNAs, lncRNAs have some unique features that make them potential diagnostic and/or prognostic candidate biomarkers in cancer and include: easy detection of lncRNAs in biological fluids, stability of lncRNAs in biological fluids and tissues, and presence of different sources of circulating lncRNAs [43]. Given that lncRNAs can be detected in body fluids such as blood, urine, and saliva, non-invasive and minimally invasive methods of cancer screening and monitoring can be implemented by employing particular lncRNAs [44] (Figure 2).

The importance of lncRNAs as tumor biomarkers lies in the following aspects.

Diagnostic Potential: Given that aberrant expression of lncRNAs is correlated with different stages of cancer development, detecting these changes can aid in the early diagnosis of cancer, enabling timely intervention and improving patient outcomes [43].

Prognostic Indicators: Certain lncRNAs are linked to disease progression, metastasis, and overall survival rates. By analyzing the expression levels of these lncRNAs, clinicians can better predict the prognoses of cancer patients and tailor treatment strategies accordingly [45,46]. For instance, the lncRNA nuclear-enriched abundant transcript 1 (*NEAT1*) has been shown to regulate immune response, and its expression level is elevated in immune-related pathologies, suggesting its potential utility as a prognostic biomarker [47].

Predictive Markers: lncRNA expression profiles can provide insights into the response of cancer cells to specific treatments. This predictive information can guide the selection of personalized treatment options, minimizing adverse effects and optimizing therapeutic outcomes [48].

For instance, the expression of lncRNAs *HULC* and *Linc00152* is significantly higher in hepatocellular carcinoma compared to normal liver tissues [49]. Similarly, prostate cancer gene 3 (*PCA3*) lncRNA has been considered a biomarker of prostate cancer [50]. In an independent study, dysregulated expression of 30 lncRNAs has been identified in non-small-cell lung cancer (*NSCLC*); therefore, these lncRNAs are potential biomarkers for non-small cell lung cancer (*NSCLC*) [51,52]. Hence, in the near future, it may become feasible to incorporate certain lncRNAs into the standard repertoire of cancer biomarkers. This anticipation arises from the recent recognition of specific lncRNAs as adaptable regulators of immune cell development, differentiation, immune modulation, and functional attributes in a cell- and context-specific manner. Notable examples encompass primate-specific lncRNAS like *FLANC* [53] and *N-BLR* [54] or lncRNA-*Cox2*, *linc1992* (also known as *THRIL*), lncRNA-*IL7R*, *HOTAIRM1*, and *lnc-DC*. From an immunological perspective, lncRNAs have been demonstrated to regulate the expression of certain genes implicated in immune responses. In this setting, several lncRNAs have been demonstrated to regulate the NF-kB, arachidonic acid, MAPK, and JAK/STAT signaling pathways, which play an essential role in orchestrating inflammation [34]. Such is the case of *NKILA* (NF-KappaB Interacting lncRNA), a cytoplasmic NF-κB interacting with long non-coding RNA that blocks IκB phosphorylation and suppresses breast cancer metastasis [55]. Furthermore, the reduction of the lncRNA *Carlr* led to a decline in the expression of NF-κB-associated genes in both mouse and human macrophages [56]. Another emerging lncRNA of significance in the context of inflammation and arachidonic acid (AA) metabolism is the extragenic RNA *PACER*, which is associated with p50 and COX-2 [57]. To summarize, the significance of lncRNAs as tumor biomarkers lies in their potential to aid in cancer diagnosis, prognosis, treatment prediction, and monitoring. Moreover, their immune-modulatory properties underline their pivotal role in shaping immune responses, offering opportunities for novel therapeutic interventions in immune-related disorders as well as cancer.

## 4. Overview of Tumor Immunity

Tumor immunity, also known as cancer immunity or anti-tumor immunity, refers to the complex interactions between the immune system and cancerous cells within the body [58]. It encompasses the body’s ability to recognize and mount a defense against cancer cells, as well as the strategies that tumors employ to evade immune surveillance. The tumor microenvironment (TME) is the area surrounding a tumor inside the body and comprising an extracellular matrix, blood vessels, fibroblasts, immune cells, inflammatory mediators, e.g., chemokines and cytokines, and others [59,60,61]. The dysregulated control of inflammation contributes to the initiation, promotion, and metastasis of cancer. Clinical studies have demonstrated that the use of anti-inflammatory drugs displays anti-cancer activity [62,63,64]. The field of tumor immunity is crucial for understanding the dynamics of cancer development, progression, and potential therapeutic interventions (Figure 2). Here, we provide an overview of the key aspects of tumor immunity.

### 4.1. Immune Response to Tumors

The immune system can recognize abnormal or foreign cells, including cancer cells, through a process called immune surveillance. Immune cells, such as T cells, B cells, natural killer (NK) cells, and macrophages, play pivotal roles in detecting and responding to tumor cells. Furthermore, cytotoxic T cells secrete interferon γ (IFN-γ) and inhibit angiogenesis and tumor progression [65]. Tumor-associated macrophages (TAMs) are very important cells inside the TME that regulate tumor proliferation, metastasis, angiogenesis, and local immune suppression. Given the plasticity of TAMs, they can be broadly classified into two major subsets, M1 and M2, which exert different actions. M1 polarized macrophages are cells that confer pro-inflammatory effects by secreting pro-inflammation cytokines that induce immune response and inhibit tumor growth. On the contrary, the M2 phenotype exhibits anti-inflammatory action and stimulates the production of anti-inflammation cytokines that suppress the immune response and thus promote tumor growth and metastasis [66]. Immune responses against tumors can involve both innate immunity (rapid, non-specific responses) and adaptive immunity (specific, memory-based responses).

### 4.2. Cancer Immune Evasion

Tumors can evolve mechanisms to evade immune detection and destruction, a phenomenon known as immune evasion. Tumor cells may downregulate the expression of antigens that immune cells recognize, making them less visible to the immune system. These cells can also create an immunosuppressive TME by secreting cytokines that inhibit immune responses [67]. Besides TAMs, myeloid-derived suppressor cells (MDSC) and regulatory T (Treg) cells are the main components of the immunosuppressive TME that induce T cell dysfunction, thereby increasing tumor progression and metastasis [66,68]. Thus, the exogenous regulation of TME, in terms of immune cells, chemokines, cytokines, miRNA, and lncRNA, has emerged as a promising strategy against cancer treatment [66,69] (Figure 2).

Immune checkpoints regulate the intensity of immune responses, preventing excessive tissue damage [67]. However, tumors can exploit these checkpoints by suppressing immune activity. Immunotherapy involves targeting immune checkpoints, such as PD-1/PD-L1 or CTLA-4 [70,71,72], to restore the immune system’s ability to recognize and attack cancer cells. Checkpoint inhibitors have shown remarkable success in treating certain types of cancers by unleashing the immune response against tumors [73,74,75,76,77,78].

Nowadays, cancer vaccines, also named treatment vaccines or therapeutic vaccines, aim to boost the immune system to recognize specific tumor antigens, training it to target and destroy cancer cells. Treatment vaccines work in different ways: they can block cancer relapse, kill cancer cells, and inhibit tumor growth and metastasis. On the other hand, adoptive T cell therapy, also named adoptive immunotherapy and immune cell therapy, involves collecting and engineering a patient’s T cells to specifically target cancer cells [79,80]. There are two main types of adoptive T cell therapy: tumor-infiltrating lymphocytes (or TIL) therapy and CAR-T cell therapy. They both use modified patient-derived T cells which are then infused back into the patient’s body. Tumor cells often express unique antigens, called tumor-associated antigens (TAAs), which can be targeted by the immune system. Neoantigens are antigens generated from tumor-specific mutations. They are unique to each patient’s tumor and have become a focus of personalized cancer immunotherapy [81]. The adaptive immune system has memory cells that “remember” previous encounters with antigens, enabling it to respond more effectively upon re-exposure. Some cancer immunotherapies aim to establish immune memory to ensure long-term control and prevention of tumor recurrence [82].

Overall, understanding the interactions between the immune system and tumor cells is essential for developing effective cancer treatments. Advances in tumor immunity research have led to groundbreaking therapies that harness the body’s immune responses to combat cancer and hold promise for improved patient outcomes in the future.

### 4.3. ncRNAs and Immune Response

The role of ncRNAs in tumorigenesis and immune response has been investigated widely in cancers. In these settings, ncRNAs regulate proliferation, differentiation, apoptosis, necrosis, autophagy, immune response, and inflammation [35,69,83,84]. In colorectal cancer, it was reported that the tumor suppressor miR-195-5p promotes TAM polarization by suppressing *NOTCH2* expression [85]. Furthermore, exosomal miRNAs (e.g., *miR-934*, *miR-25-3p*, *miR-130b-3p*, *miR-425-5p*) can induce the activation of the CXCL13/CXCR5 or CXCL12/CXCR4 axis in colorectal cancer cells, which in turn activate TAM polarization and metastasis to liver [86]. On the other hand, lncRNAs such as *lnc-EGFR*, *SNHG1*, *Flicr*, and *Flatr* can orchestrate the correct function and differentiation of Treg cells [86]. Moreover, lnc-*EGFR* has also been shown to stimulate Treg differentiation, inhibit cytotoxic T lymphocyte activity, and induce hepatocellular carcinoma (HCC) growth [87]. In contrast, *NIFK-AS1* lncRNA inhibit the M2-like polarization of macrophages, proliferation, migration, and invasion of endometrial cancer, at least in part, by inhibiting *miR-146a* [88]. The process of immune evasion is primarily driven by the establishment of an immunosuppressive environment, a phenomenon that can be orchestrated by long non-coding RNAs (lncRNAs). Certain lncRNAs even facilitate the development of resistance to treatments via the PD-1/PD-L1 pathway and the presentation of inhibitory antigens. For instance, the lncRNA *MALAT1* is capable of modulating tumor immunity by indirectly enhancing the expression of PD-L1 through its interaction with *miR-195* and *miR-200a-3* [45,89]. Additional information on lncRNA can be found in the following sections. The cellular communications between the different components in the TME are undoubtedly complex; however, during the last 10 years, our knowledge of the role of immune response in regulating tumor development has increased drastically.

In the current landscape, the idea of leveraging cellular reprogramming to strategically reshape an adverse immune response or reshape tumor attributes from a detrimental condition to a favorable one that actively reinforces the battle against tumors is a relatively unexplored frontier. Nonetheless, the prospect of turning this vision into tangible reality lies tantalizingly close, propelled by remarkable strides in genetic profiling and cutting-edge technology (See Figure 2). In this context, this review underscores the critical significance of not only eradicating malignant cells and thwarting drug resistance but also achieving these goals while preventing the emergence of an unfavorable TME. Harnessing the inherent potential of the immune system’s modulation will have a pivotal role in accomplishing this goal.

## 5. Immune Response and Long Non-Coding RNAs

In recent years, the intricate regulatory roles of long non-coding RNAs (lncRNAs) in the immune response have gained substantial attention in the field of molecular biology. These RNA molecules, which do not code for proteins but are crucial in gene regulation, have emerged as influential modulators of immune cell differentiation, activation, and inflammatory processes.

lncRNAs play a pivotal role in steering the fate of immune cells. They fine-tune the differentiation of various immune cell types, including T cells, B cells, macrophages, and dendritic cells, by regulating gene expression patterns. Additionally, lncRNAs are instrumental in driving immune cell activation, dictating the intensity and duration of immune responses [33]. In the context of inflammatory responses, lncRNAs act as critical regulators, impacting the production of key immune mediators such as cytokines and chemokines. Some lncRNAs amplify inflammatory signals, while others exert suppressive effects, contributing to the dynamic equilibrium of immune reactions. This dual role places lncRNAs at the center of maintaining immune homeostasis [90].

Cancer immunology has unveiled a novel dimension of lncRNAs’ significance. Within the tumor microenvironment, lncRNAs influence immune cell infiltration, immune evasion mechanisms, and the establishment of an immunosuppressive milieu. These molecules actively shape the intricate interplay between tumor cells and immune responses, impacting disease progression and therapeutic outcomes [91]. The remarkable immunomodulatory roles of lncRNAs hold promise for therapeutic interventions. As lncRNAs impact immune checkpoint molecules, cytokines, and other immune regulators, they offer potential targets for novel immunotherapies. Manipulating their expression levels could potentially enhance the efficacy of immune checkpoint blockade, adoptive T cell therapies, and other immune-based treatments.

Recent research findings have additionally reported lncRNAs participate in cancer onset and progression via reprogramming the tumor immune microenvironment (TIME). Certainly, the overexpression of specific lncRNAs is strongly linked to the infiltration of immune cells and serves as a prognostic indicator for cancer patients. For example, lncRNA-*LINC00665* is associated with the infiltration rates of macrophages and dendritic cells (DCs). It also has a role in inhibiting regulatory T cells (Tregs) and averting T cell exhaustion by functioning as a competing endogenous RNA (*ceRNA*), along with *FTX* [92]. *lncRNA-TCL6* exhibits a direct correlation with the infiltration of tumor-infiltrating lymphocytes (TILs) and the presence of immune checkpoint proteins such as PD-1, PD-L1, and CTLA-4 [93]. The oncogenic lncRNA *LINK-A* diminishes the ability of tumor cells to present antigens, thereby undermining immune surveillance. This phenomenon contributes to the evasion of cancer cells from immune checkpoints and supports the survival of malignant cells [94].

The stimulation of the immune system to recognize and attack tumor cells is an attractive means of facilitating the complete elimination of tumors [95]. It has been observed that infiltrating cytotoxic T cells (CTLs) can be localized in tumor sites; however, these germinal cells are only present in a small proportion. T cells probably lack either distinctive antigenic peptides or the cell adhesion or co-stimulatory molecules necessary to elicit a correct primary T cell response [96]. The difficulty to induce an effective anti-tumor immune response largely stems from the highly immunosuppressive microenvironment present in tumors and thus far, no effective immunostimulatory strategies have been developed to effectively enhance CTL activity [96,97]. By targeting specific lncRNAs, it becomes plausible to disrupt the immunosuppressive networks within tumors. This disruption could potentially create an environment where cytotoxic T lymphocytes (CTLs) are capable of infiltrating and effectively eliminating tumor cells. In an extensive research paper featured in Nature Immunology, a recently discovered long non-coding RNA (lncRNA) called nuclear factor-κB (NF-κB)-interacting lncRNA (*NKILA*) was demonstrated to have a specific role in making anti-tumor T cells more susceptible to cell death upon stimulation by tumor-related antigens. The study revealed that when *NKILA* is suppressed, it leads to an increased infiltration of cytotoxic T lymphocytes (CTLs) and a reduction in tumor growth. Consequently, this discovery presents a potential strategy to amplify the efficacy of T cells in adoptive T cell therapy for cancer [98].

lncRNAs can simultaneously regulate multiple pathways, which can prevent pathway redundancy or resistance, a feature which cannot be achieved by many other therapeutic agents; therefore, they may serve as promising agents for enhancing CTL function in several tumors [99].

## 6. lncRNAs and Innate Immune Response

The innate immune system serves as the initial defense line against infections, engaging in the identification and elimination of harmful agents. The primary recognition of ubiquitous pathogenic components, encompassing pathogen-associated molecular patterns (PAMPs) and damage-associated molecular patterns (DAMPs), occurs through pattern recognition receptors (PRRs) such as toll-like receptors (TLRs), nucleotide binding and oligomerization domain (NOD)-like receptors (NLRs), RIG-I-like receptors (RLRs), and C-type lectin receptors (CLRs). Consequently, this recognition triggers the subsequent activation of key factors like the inflammasome and diverse transcription factors including nuclear factor-κB (NF-κB) and interferon response factors (IRFs). These factors collectively orchestrate the inflammatory response essential for eradicating pathogens [100]. These receptors are present on myeloid cells affiliated with the innate immune system, encompassing monocytes, macrophages, and dendritic cells. Additionally, they are found on tissue-associated cells such as epithelial cells and fibroblasts [101]. Long non-coding RNAs (lncRNAs) are currently implicated in various tasks in the innate immune response. These encompass tasks like preserving hematopoietic stem cells; directing the differentiation and programmed cell death of myeloid cells; and promoting the activation of monocytes, macrophages, and dendritic cells. In initial studies, the differential regulation of lncRNAs in innate immune responses was reported in virus-infected cells (SARS-CoV) and dendritic cells (DC) after stimulation with lipopolysaccharide (LPS), which is par excellence the agonist of TLR4 [102,103]. In this sense, studies show the innate immune responses mediated by TLRs have a promising biological role in cancer regulation [104].

TLRs are usually involved in the recognition of specific pathogen-associated molecular patterns (PAMPs) derived from bacteria, viruses, fungi, and protozoa. The activation of the receptors allows a coordinated immune response to clear infection and eliminate the pathogens [105,106]. Likewise, TLRs play a crucial role in tissue homeostasis by regulating wound healing, non-infectious inflammation, and tissue regeneration [107,108]. The regulation of inflammatory mediators (e.g., cytokines, acute proteins, and anti-microbial peptides, among others) by lncRNAs via TLRs is not fully understood [109,110]. However, some studies reported that TLR4 and TLR7/8 can be activated by specific agonists (e.g., LPS, and R848 synthetic anti-viral compound) that increase the expression of lncRNA-*Cox2* via the MyD88-NFkB pathway [34]. lncRNA-Cox2 showed nuclear and cytosolic localization in murine bone marrow-derived macrophages (BMDMs) [111]; the regulation of lncRNA-Cox2 participated both in the activation and repression of immune responses. Moreover, the complex of lncRNA-*Cox2* with *hnRNP-A/B* and *hnRNP-A2/B1* can regulate the repression of CCL5 [109,112]. This immune mediator participates in the recruitment of T cells, eosinophils, neutrophils, and basophils to the inflammatory site. Furthermore, in gastric cancer, CCL5 levels correlate with tumor progression and prognosis, whereas in breast cancer, CCL5 produced by breast cancer cells increases the production of matrix metalloproteinase by T cells and/or monocytes. Interestingly, systemic treatment of mice with neutralizing anti-CCL5 antibodies reduced the extent of subcutaneous tumors, liver metastases, and peritoneal carcinosis. In a similar context, the knockdown of CCL5 from CT26 (mouse colon tumor cells) inhibited apoptosis of CD8+ and consequently reduced the size of the tumor in the mice model [113,114].

For example, this inflammatory chemokine activates NK cells. It is crucial to note that NK cells exhibit swift and robust anti-tumor immunity, rendering this potential therapy a focal point of investigation in clinical settings [114]. Considering these dynamics, there is a plausible avenue to explore therapies involving the lncRNA-Cox2. By harnessing its potential, cytokine production could be modulated, potentially thwarting or mitigating cancer progression via the regulation of immune cell activity.

Yet, in the human monocyte cell line (THP-1), it was reported that a positive and negative feedback system produced tumor necrosis factor-alpha (TNF-α) and interleukin 6 (IL-6) via TLR2, and this production was found to be regulated by the lncRNA *linc1992/THRIL* [115]. This regulation was analyzed by a pull-down assay and an RNA immunoprecipitation (RIP) assay, where *linc1992* and *hnRNPL* formed an RNP complex in vivo. Moreover, chromatin immunoprecipitation (ChIP) assay revealed that hnRNPL binds to the TNF-α promoter region, while the knockdown of *linc1992* showed a reduced binding of hnRNPL to the TNF-α promoter region [115]. TNF-α cytokines have potent anti-tumoral properties, and, as the name implies, cause cancer cell death [116]. TNF can be an endogenous tumor promoter because TNF stimulates tumor growth, proliferation, invasion, metastasis, and angiogenesis. However, TNF can also be a cancer killer, and its anti-tumor role may involve immune responses, e.g., promoting tumor stromal destruction by CTL or tumor-infiltrating macrophages [117]. However, TNF-α could stimulate proliferation, survival, migration, and angiogenesis in most cancer cells, resulting in tumor promotion. In sum, TNF-α plays the role of a double-edged sword that could be either pro- or anti-tumorigenic [116].

In addition, an lncRNA microarray analysis showed that the stimulation of THP-1 cells with a TLR4 agonist (i.e., lipopolysaccharide) induces the expression of almost 443 lncRNAs by more than twofold and decreases the expression of 718 lncRNAs, which is double the number of both increased and decreased lncRNAs compared to cells treated without the agonist [112]. In human peripheral blood mononuclear cells, lnc-*IL7R* was one of the most upregulated lncRNAs after TLR4 and TLR3 activation by lipopolysaccharide and Pam3CSK4, respectively [118]. In addition, negative expression of E-selectin, VCAM-1, IL-8, and IL-6 was observed following TLR4 stimulation [118]. The mechanism by which lncRNA-*IL7R* regulated E-selectin and VCAM-1 is dependent on the trimethylation of histone H3 at lysine 27 (H3K27me3) [118]. Moreover, increased plasma levels of VCAM-1 and E-selectin are associated with the advanced stage of breast cancer and with the presence of circulating cancer cells [119]. Activation of different TLRs in cancer cells results in an inflammatory response that promotes tumorigenesis; however, TLR has also been found to induce strong anti-tumor activity by indirectly activating the tolerant host immune system to destroy cancer cells [104]. Therefore, the specific modulation of TLRs by a novel or typical agonist could stimulate the expression of specific lncRNAs that participate in immune checkpoints. Increasing the expression of these lncRNAs is a promising new strategy to treat cancer cells. More examples of ncRNA and the regulation of innate immune response were reviewed in detail in a recently published extensive review [34].

## 7. Cell-Mediated Immunity and lncRNAs

Immune cells have a direct impact on the trajectory of cancer progression or resolution [7]. Yet, the potential of using lncRNAs to orchestrate lineage differentiation as a therapeutic avenue against cancer remains unexplored. A remarkable feature of the expression of lncRNAs is that they are expressed across a spectrum of immune cells, spanning monocytes, macrophages, dendritic cells, neutrophils, T cells, and B cells. This expression intricately corresponds to various immune cellular stages [7]. lncRNA expression is prompted by stimuli encompassing development, differentiation, activation, and immune responses, mediated through diverse mechanisms such as dosage compensation, imprinting, enhancer functions, and transcriptional regulation [33,120] (Figure 2).

Under normal circumstances, immune cells undergo differentiation within the myeloid and lymphoid lineages. However, during infectious or inflammatory processes, immune cells can undergo differentiation [121,122]. Recent investigations have unveiled that a diverse group of lncRNAs participate in B cell differentiation stages, spanning from pre-B1 to memory cells, including plasma blast cells derived from human bone marrow, naive, and memory cells. This comprehensive meta-analysis serves as a foundational step toward identifying lncRNAs associated with malignant lymphomas originating from distinct normal B cell stages, specifically from germinal centers [123].

In the context of inflammatory responses and infectious diseases, hematopoietic differentiation can determine the clearance or progression of infections and inflammatory processes [124]. Noteworthy among the lncRNAs that are implicated in such processes is the lncRNA *HOTAIRM1*. Particularly, *HOTAIRM1* significantly contributes to hematopoietic cell differentiation and is highly expressed during induced granulocytic differentiation in the NB4 promyelocytic leukemia cell line and neutrophils [125]. Neutrophils, while classically involved in anti-microbial functions, also exert substantial influence on the tumor microenvironment. Through the production of cytokines and chemokines, neutrophils facilitate inflammatory cell recruitment and activation. Additionally, their release of reactive oxygen species and proteinases plays a pivotal role in regulating tumor cell proliferation, angiogenesis, and metastasis [126]. Interestingly, studies have linked infiltrating neutrophils in bronchoalveolar carcinoma, melanoma, renal carcinoma, and head and neck squamous-cell carcinoma (HNSCC) to a poor prognosis. As we explore inducing or suppressing neutrophil differentiation as a cancer strategy, lncRNAs are promising targets in this context.

Dendritic cells (DCs), renowned as “nature’s adjuvants”, wield the ability to govern immune tolerance and active immunity. These cells serve as natural vehicles for antigen delivery and exert immunologic effects on tumors [127]. In mouse models, DCs exhibit the capacity to capture tumor antigens released by tumor cells, presenting these antigens to T cells in tumor-draining lymph nodes. This presentation results in the generation of tumor-specific cytotoxic T lymphocytes (CTLs) that contribute to tumor rejection. As such, DCs present themselves as pivotal targets for therapeutic interventions against cancer. Recent discoveries highlight the upregulation of the lncRNA *lnc-DC* during human DC development. Specifically, lnc-DC engages with STAT3, maintaining its phosphorylation [128]. Intriguingly, the functional consequences of *lnc-DC* knockdown encompass impaired expression of membrane receptors vital for T cell activation—CD80/86, HLA-DR, and CD40—as well as compromised antigen presentation and reduced IL-12 production post stimulation [129]. Unraveling the intricate role of lnc-DC in dendritic cell differentiation requires further investigation. The alignment between lncRNA regulation and immune cell differentiation may be an innovative strategy for defining cancer progression. Overall, the interplay between lncRNAs and immune cell dynamics provides a novel avenue for understanding and potentially influencing the intricate landscape of cancer progression.

## 8. Adaptive Immunity

The realm of adaptive immunity traditionally encompasses B and T cells. However, it is important to recognize that ncRNAs have substantial regulatory influence over lymphocyte biology, including interactions with vital pathways such as NF-κB, NOTCH, MYC, and TCR/CRR signaling. Moreover, lncRNAs hold relevance in shaping cell effector functions, underscoring their intricate involvement [34]. Several lncRNAs have emerged as key players in modulating adaptive immunity. These include *LincrR-Ccr2-5AS*, *GAS5*, and *NeST*, which intricately govern TH2 cells, TH17 cells, and CD8+ cells, respectively.

*LincrR-Ccr2-5AS* is prominent in intergenic lncRNA expression and regulation during T cell development and differentiation. The associated gene significantly impacts the chemokine-mediated signaling pathway, meaning that it has a pivotal role in cell differentiation and migration. Notably, in lymphocytes with suppressed *LincrR-Ccr2-5AS* expression, Ccr1 was downregulated compared to Ccr5, both of which are crucial chemokine receptors for migration. This intricate connection is further supported by the finding that CD45+ cells with reduced LincrR-Ccr2-5AS expression demonstrated impaired lung migration in a murine model [130].

Maintaining proper lymphocyte population proliferation is essential for immune response regulation and the prevention of leukemic and autoimmune diseases. Within this context, growth-arrest-specific transcript 5 (*GAS5*) emerges as a critical factor for normal growth arrest in T cells and undifferentiated lymphocytes. Intriguingly, overexpression of GAS5 triggers increased apoptosis rates and a decline in cell-cycle progression, ultimately resulting in a significant reduction in lymphocyte populations. Thus, the potential therapeutic application of targeting *GAS5* holds promise in the realm of leukemia and other cancer types [131].

NeST (nettoie Salmonella pas Theiler’s) is intriguing because of its interaction with WDR5, a histone component of the H3 lysine 4 methyltransferase complexes. This interaction leads to the alteration of histone 3 methylation at the IFN-γ locus. The implications of this interaction are profound, as this lncRNA influence extends to epigenetically regulating IFN, subsequently impacting susceptibility to viral and bacterial pathogens. The implications of NeST’s actions have implications for dysregulated IFN-γ activity in human disease, including cancer [132].

Innate and adaptive responses converge in a sophisticated network, collaborating to effectively function and combat pathogens and tumor cells. A delicate balance within this communication network is paramount for accurate immune function. Minor deviations can disrupt this intricate balance, leading to altered immune functions. The ongoing exploration of lncRNAs associated with adaptive responses holds the promise of unveiling novel insights into the landscape of cancer.

## 9. Harnessing lncRNAs: A New Dimension in Immune Checkpoint Regulation

Cancer cells employ a multifaceted approach to evade immune surveillance, with the modulation of immune checkpoint molecules emerging as a significant mechanism of immune evasion. This section explores the intricate relationship between immune checkpoints and long non-coding RNAs (lncRNAs) within the context of cancer immunotherapy.

### 9.1. Checkpoint Manipulation: A Strategy for Immune Evasion

Cancer cells often manipulate immune checkpoint molecules to thwart their elimination by immune cells. While experimental use of immune checkpoint inhibitors (ICI) has shown promise, its clinical translation has been met with limited success. Conversely, the exploration of agonists for stimulatory immune checkpoints is underway [133,134]. Stopping the activation of inhibitory immunoreceptors has potential in the reactivation of anti-tumoral immune functions, a concept currently under experimental scrutiny and poised for future clinical application. Several cancer-related inhibitory immunoreceptors related to cancer have been identified, including PD-1 (programmed cell death protein 1), CTLA-4 (T lymphocyte-associated antigen), LAG-3 (lymphocyte-activation gene 3), TIM-3 (T cell immunoglobulin domain and mucin domain-3), TIGIT(T cell immune receptor with immunoglobulin and ITIM domain), and BTLA (B and T lymphocyte attenuator) [133].

Programmed death protein 1 (PD-1) and its ligand 1 (PD-L1) exemplify the pivotal role of negative checkpoints in cancer immune evasion. PD-1, expressed on the surface of T cells and myeloid cells, engages PD-L1, which is present on tumor cells [135]. This interaction effectively dampens T cell proliferation, cytokine production, and cytotoxic activity [135]. Antibodies targeting PD-L1 have shown clinical success in diverse cancers, including melanoma, leukemia, lymphoma, liver cancer, colorectal cancer, urothelial cancer, squamous-cell carcinoma of the head and neck, cervical cancer, kidney cancer, stomach cancer, and breast cancer [136,137]. While monotherapy and combination therapies have demonstrated clinical promise, certain cancers exhibit a low response rate, primarily due to the absence of well-defined biomarkers, issues of toxicity, and drug resistance [137].

### 9.2. lncRNAs Enter the Arena: A Novel Perspective

Recent revelations have unveiled a new role for lncRNAs in cancer progression, namely lncRNAs that could become potentially targetable PD-L1 inhibitors. For instance, Qu, Shuang, et al. discovered an IFNγ-upregulated splice isoform of the PD-L1 lncRNA that enhances the proliferation and invasion of lung adenocarcinoma cells through direct binding to c-Myc, enhancing its transcriptional activity. This groundbreaking discovery underscores the potential of combined PD-L1 and *PD-L1-*lncRNA therapies not only in lung cancer but also in various cancer types [138].

Beyond PD-1/PD-L1 pathways, lncRNAs, such as lnc-*OC1*, have emerged as regulators of PD-L1 expression. In endometrial cancer cells, lnc-*OC1* enhances PD-L1 expression, bolstering cell viability and thwarting apoptosis [139]. Similarly, several lncRNAs positively correlate with PD-L1 in hepatocellular carcinomas (HCCs), including *MIR155HG*, *PCED1B-AS1*, and *MIAT* [140]. Notably, *MIAT* strongly correlates with the expression of PD-1, PD-L1, and CTLA4 in HCC, influencing immune escape mechanisms and resistance to sorafenib [140]. Over the past 5 years, research groups have diligently pursued the identification of immune-related lncRNA signatures across diverse cancers. These signatures encompass various immune factors, such as immune cells, cytokines, and immune checkpoints, all of which play pivotal roles in the progression and regulation of the tumor microenvironment. Importantly, the activity of these factors can be modulated by lncRNAs (see Table 1), underscoring the potential of targeting specific lncRNAs to curb cancer cell proliferation and enhance the efficacy of immunotherapies.

## 10. Navigating the Challenges: Targeting lncRNAs for Immunotherapy

With their remarkable capabilities, lncRNAs are risen as promising new targets that influence the treatment of various diseases, including cancer. Hence, they can play a significant role in personalized cancer therapy.

### 10.1. Intricate Cellular Localization

Such targeting approaches seek to reduce the impact of oncogenic lncRNAs or disrupt their functions to thwart the development of cancer. Several strategies, such as suppressing oncogenic lncRNAs, modifying their epigenetic influence, interfering with their activity, reinstating downregulated or lost lncRNAs, and harnessing regulatory elements and expression patterns of lncRNAs, have been proposed for therapeutic targeting of lncRNAs in cancer. These methods have demonstrated inhibitory effects on malignant processes [142]. In the case of maintaining the effectiveness of lncRNAs in performing their biological functions, it is crucial that the transfer of these lncRNA molecules preserve their secondary structure. Natural intercellular transfer of lncRNAs occurs through extracellular vesicles (EVs), and numerous research efforts have demonstrated that delivering lncRNAs via EVs contributes to disease progression across nearly all organ systems [143,144,145]. The potential of long non-coding RNAs (lncRNAs) in immunotherapy is undeniable, given their multifaceted roles in gene regulation, encompassing epigenetic, transcriptional, and post-transcriptional activities. An extensive review about the role of lncRNA in tumor immunotherapy published by Pan et al. in 2023 can be revised in detail to explore this area of research [91,146]. However, harnessing lncRNAs for therapeutic benefit presents several formidable challenges.

lncRNAs exhibit diverse localization patterns within cells, residing both in the cytoplasm and nucleus. Furthermore, these molecules are secreted in exosomes, facilitating intercellular communication (Figure 3). In the context of the TME, exosomal lncRNAs play critical roles in cancer cell proliferation, immunosuppression, and chemoresistance [147]. The challenge here lies in delivering specific lncRNAs to their intended targets. lncRNAs are vulnerable to rapid degradation by nucleases in biological fluids. Additionally, ensuring their selective delivery to target cells remains a daunting task. Achieving subsequent activity within the cytoplasm or nucleus complicates the design of lncRNA-based therapeutic strategies for cancer treatment. Numerous approaches have been explored to tackle these challenges [148].

### 10.2. Delivery Strategies

Lentiviral vectors, although used in experimental settings to achieve targeted lncRNA overexpression or knockdown, pose concerns regarding their safety due to the theoretical risk of oncogene insertion [149]. Third-generation lentiviral vectors have shown promise in clinical trials for gene introduction in hematopoietic stem cells and CAR T cell therapy [149].

Nanoliposomes have shown promise in preclinical studies [39,150]. However, optimizing their composition for efficient nuclear delivery remains a continuous challenge. Innovative combinations of liposomes and lipid nanoparticles (NPs), coupled with aptamers, antibodies, peptides, and protein ligands, have been explored for active or passive targeted drug delivery systems [149]

Uncertainties and Risks. Addressing the stability and appropriate dosages of lncRNAs, as well as their immunogenicity, administration routes, real risks, and potential side effects is paramount. Achieving this requires a deep understanding of lncRNAs and their intricate regulation. While the Food and Drug Administration (FDA) and European Medicines Agency (EMA) have approved nearly 14 types of liposomal products for various applications, including vaccine adjuvants and drug delivery [151], careful consideration of strategies such as neutral DOPC-based nanoliposomes is necessary for successful in vivo trials [152].

In conclusion, the journey to harnessing lncRNAs for resolving chronic and acute diseases, including cancer, involves addressing these multifaceted challenges. Experimental and clinical trials will continue to pave the way toward discovering the most effective strategies for lncRNA delivery, ultimately advancing the field of immunotherapy.

## 11. Conclusion and Perspectives

lncRNAs have emerged as versatile regulators with essential roles in immune modulation, cellular physiology, development, and the pathogenesis of diseases, including cancer. Their multifaceted functions underscore their potential as valuable therapeutic targets in cancer treatment. However, it is evident that the therapeutic targeting of lncRNAs, along with the efficient and safe delivery of therapeutic agents, has been relatively overlooked compared to that of their RNA counterparts such as mRNA and microRNAs. Notably, the limited number of clinical trials involving lncRNAs, exemplified by a single reported trial (www.clinicaltrials.gov, NCT02641847, accessed on 1 January 2020) [153], highlights the substantial gaps that persist in this evolving field. To catalyze progress in the development of lncRNA-targeting therapeutics, it is paramount to confront the formidable challenges that lie ahead.

Exploiting bioinformatics, comprehensive databases, and high-throughput technologies is fundamental. This approach can unravel critical insights into various facets of lncRNAs, such as their localization, intricate structural features, functional motifs, underlying mechanisms of action, and intricate interactions with other vital biological molecules. A comprehensive functional screening strategy emerges as a necessary step to pinpoint suitable lncRNAs as therapeutic targets.

Delving into the properties of modified oligonucleotides is indispensable. A profound understanding of these properties is essential not only to mitigate potential toxicity concerns but also to facilitate the development of efficient and safe drugs targeting lncRNAs.

Beyond these pivotal steps, several other dimensions must be explored in the journey of lncRNA-targeted therapeutics.

Innovative delivery systems must be developed to ensure the precise and effective transport of therapeutic agents to their intended lncRNA targets. Achieving this level of precision is crucial to optimize therapeutic outcomes.

Recognizing that the roles of lncRNAs can vary across different cancer types and stages is vital. Tailoring therapeutic strategies to specific contexts can enhance their efficacy.

Given that lncRNAs operate within intricate cellular regulatory networks, gaining a comprehensive understanding of their specific functions within these networks is essential. This knowledge can unveil novel therapeutic avenues.

Strategies to ensure the stability and bioavailability of therapeutic agents warrant thorough exploration. These efforts are pivotal in maximizing the therapeutic potential of lncRNA-targeted drugs.

In summary, the potential of lncRNAs as therapeutic targets in cancer treatment is immense. Yet, to harness this potential effectively, several multifaceted challenges, ranging from the need for a deeper understanding of lncRNA biology to the development of innovative delivery systems and safety considerations, must be confronted. As we navigate the intricate landscape of lncRNAs, continued innovation and collaborative endeavors will be instrumental in unlocking their full therapeutic promise, ultimately advancing the field of lncRNA-targeted cancer therapeutics.

## Figures and Tables

**Figure 1 ncrna-09-00070-f001:**
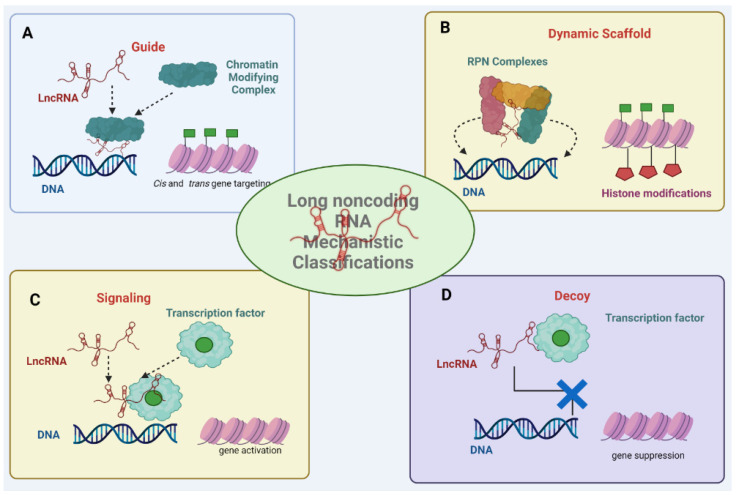
Fundamental mechanisms involved in categorizing lncRNAs. (**A**) lncRNAs function as guides directing chromatin-modifying complexes to precise genomic sites to regulate gene expression. (**B**) lncRNAs serve as flexible scaffolds facilitating the temporary assembly of cofactors. (**C**) lncRNAs act as signals by functioning as indicators or markers for distinct cellular processes. (**D**) lncRNAs function as decoys, binding to microRNAs or transcription factors, diverting them from their intended targets, and influencing both transcription and translation processes.

**Figure 2 ncrna-09-00070-f002:**
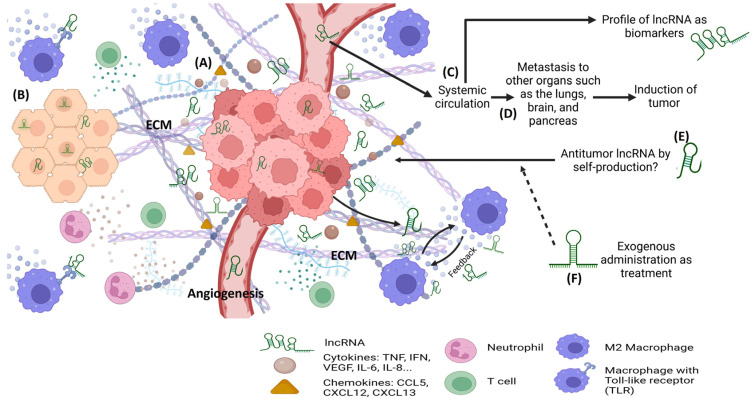
Long non-coding RNA in the tumor microenvironment. (**A**) Tumor cells produce different types of molecules, including cytokines, e.g., tumor necrosis factor α (TNF-α), interferon-γ (IFN- γ), vascular endothelial growth factor (VEGF), interleukin 6 (IL-6), interleukin 8 (IL-8), chemokines (e.g., chemokine ligand 5 and stromal cell-derived factors 12 and 13), the extracellular matrix (ECM), and lncRNAs, which promote angiogenesis and changes in the phenotype of organ cells and dysregulated cell proliferation. (**B**) Healthy adjacent cells receive and recognize the cell signaling of lncRNA in cytoplasm and nucleus. Additionally, the cellular immune system releases soluble mediators. Taken together, these phenomena induce changes and transformation to tumor phenotype cells. (**C**) The analysis of systemic circulation helps identify specific types of lncRNA that can be used as biomarkers and predictive molecules for cancer status. (**D**) Moreover, some lncRNA could be related to metastasis and tumor inducers. (**E**) It is possible that naturally circulating lncRNA produced by the immune system, stem cells, or another source can participate in tumor control. (**F**) Thus, the exogenous administration of specific lncRNA mimics or ncRNA targeting lncRNAs can be used as a strategy to control angiogenesis, cell proliferation, inhibition of cell differentiation, regulation of ECM production, and regulation of immune cells.

**Figure 3 ncrna-09-00070-f003:**
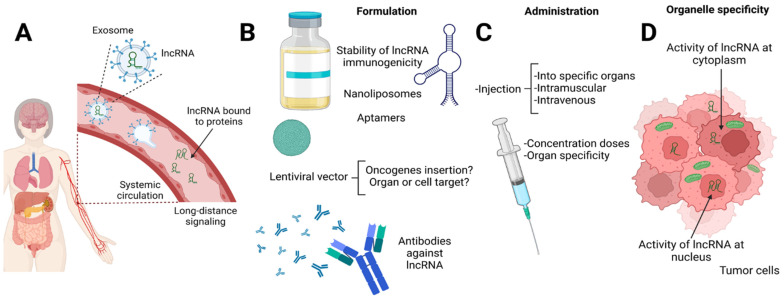
Delivery methods and challenges of lncRNA-based therapies. (**A**) lncRNAs are transported to distal organs by systemic circulation into exosomes and microvesicles and are bound to proteins. These lncRNAs could participate in promoting tumors in other organs or as a mechanism to control the tumor microenvironment. (**B**) The major challenge of using lncRNA therapies is the need for a perfectly designed formulation of carriers or strategies for the delivery of lncRNA. Whatever the strategy (e.g., lentiviral vectors, nanoliposomes, aptamers), it must maintain the stability of the lncRNA and not produce immune reactivity. Moreover, the synthesis and administration of antibodies against lncRNA related to tumor development must not recognize proteins or molecules from the patient. (**C**) Regarding administration, the correct route needs to be determined, and it is necessary to determine the standardized doses. (**D**) Formulation and administration are essential to the successfully delivery of lncRNA into the cells at the cytoplasmic, mitochondrial, or nuclear level.

**Table 1 ncrna-09-00070-t001:** lncRNA and immune regulation.

lncRNA	Cell Target and/or Effect	Regulation or Cellular Pathway	Ref.
lnc-*EGFR*, lncRNA *SNHG1*, *Flicr*, and *Flatr*	Tregs	Correct function and differentiation	[86]
lnc-*EGFR*	Treg and cytotoxic T lymphocytes	Stimulation of Treg differentiation, inhibition of cytotoxic T lymphocyte activity	[87]
*NIFK-AS1* lncRNA	Macrophages and endometrial cells	Inhibition of M2-like polarization, proliferation, migration, and invasion of endometrial cancer	[64]
*MALAT1*	PD-L1	Upregulation of PD-L1 through miR-195 and miR-200a-3	[45]
*LIMIT*	Stimulation of MHC-I and MHC-II expression	Promotes response of T cell-mediated tumor immune response	[141]
A complex composed of lncRNA-*Cox2* with hnRNP-A/B and hnRNP-A2/B1	Macrophages and repression of CCL5	Recruitment of T cells, eosinophils, neutrophils, and basophils	[112]
lncRNA *linc1992/THRIL*	Monocytes	Regulation of TNF-α	[115]
*HOTAIRM1*	NB4 promyelocytic leukemia cell line and neutrophils	Granulocytic differentiation	[125]
*lnc-DC*	Dendritic cells	STAT3 phosphorylation; deficient expression of CD80/86, HLA-DR, and CD40, impairment of antigen presentation, and decreased IL-12 production	[129]
*lincrR-Ccr2-5AS*,	TH2 cells	Cell differentiation and migration	[130]
GAS5	TH17	Apoptosis and cell-cycle progression	[131]
*NeST*	CD8+ cells (WDR5, histone component)	Production of IFN	[132]
*PD-L1* lncRNA splice isoform	c-Myc	Proliferation and invasion	[138]
lnc-*OC1*	Enhances PD-L1 expression	Apoptosis	[139]
lnc-*IL7R*	Monocytes	Regulation of E-selectin and VCAM-1 via H3K27me3	[118]

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
