# Peer review of "An Overview of the Immune Modulatory Properties of Long Non-Coding RNAs and Their Potential Use as Therapeutic Targets in Cancer"

_ncrna, 2023, doi:10.3390/ncrna9060070_

Round 1

Reviewer 1 Report

Comments and Suggestions for Authors

In the current manuscript, Moises et al. aim to present an overview about long non-coding RNAs and their immune modulatory functions in cancer. To make a long story short: This is a poorly written article that lacks a structure, especially in the Abstract and Introduction, and urgently needs thorough language editing to improve readability and allow reader to get an idea of what the authors want to convey. In its current form, this article is not publishable! After reading it, I rather feel confused and do not take away any meaningful information to remember.

As there are several problems with this manuscript, I only highlight some of them here and strongly recommend a complete makeover and re-writing:

1) Abstract

The abstract is full of mistakes and many sentences don’t even make sense. Some examples are:

The noncoding RNAs (ncRNAs) are small transcript sequences…

à Why are non-coding RNAs small? The authors themselves introduce also long non-coding RNAs.

...the variability of switch cell differentiation and activation of the different immune cell lineages showed differential regulation by the type of cancer.

à What are “switch cells”?

à Sentence needs re-phrasing

The use of lncRNA to turn off the tumor immune response and activate the anti-tumoral immune response is day by day a promising reality for the control of cancer.

à What do the authors mean with “day by day a promising reality”?

However, is imperatively elucidated the role of each ncRNA identified as well as its respective molecular pathways regulated, including the immune checkpoints, before its application at the clinical level.

à needs re-phrasing

2) Introduction

Especially the first part of the introduction needs re-writing and better punctuation. In addition, some sentences contain misleading information, e.g.:

The endogenous small ncRNA comprises mainly: small transfer RNA (tRNA), ribosomal RNA (rRNA), small interfering RNA (siRNA), and microRNA (miRNA).

à I would argue that 18S und 28S rRNA are both rather long ncRNAs

3) Overview of tumor immunity

This part lacks a clear structure. The authors jump from one topic and example to the other and don’t really refer to the heading of this paragraph. Again, several sentences don’t really make sense or fit in or don’t provide sufficient information for the reader to understand the mechanism, e.g.:

…where the regulation is by the stage of the disease and the organ affected (solid or not solid tumor)

Furthermore, the exogenous regulation of several microenvironmental factors has been emerging as a promising strategy against cancer treatment (27).

The formation of an immunosuppressive microenvironment, has been observed by the activity of lncRNAs over immune check points, like as lncRNA, MALAT1, that can upregulate the expression of PD-L1 through miR-195 and miR-200a-3, respectively (34).

Nowadays, the concept of switching the negative immune response or tumor profile to an antitumor profile is latent, and possibly it will be a reality shortly.

Nevertheless, is imperative a continuing study of the molecular and cellular pathways regulated and their clinical implications.

Here, we emphasize the importance to eliminate the malignant cells to abrogate the drug resistance and their elimination without giving rise to an adverse microenvironment, using the owner immune modulation.

4) Role of lncRNAs in cancer

This part lists random examples of ncRNAs and their described function in tumor cells and has nothing to do with the focus of this review. Personally, I would suggest to remove this section.

Furthermore, several sentences, like before, don’t make sense and some even create misleading connections. For example:

Cancer is the second leading cause of death globally which causes approximately 10 million in 2020 (35). à Causes what? Sentence is incomplete

Furthermore, some lncRNAs can act as oncogenes whereas others can act like tumor suppressor genes and this feature renders lncRNAs as potential diagnostic/prognostic biomarkers in cancers (38-40). However, this is not an easy task, because this requires considering the correct identification, stratification, patient personalization, drug delivery, and toxicity of the lncRNA (1, 3, 41).

à several false logical connections between and within both sentences

5) LncRNAs as tumor biomarkers

Again, a very generic section that can be found in every review about ncRNAs. What is the connection to the topic / title of this particular review?

6) Activating antitumor immunity using long non-coding RNAs

This section does not contain any meaningful information. In fact, the content does not reflect the heading.

7) LncRNAs and innate immune response

This section needs a better organization and language editing, e.g.:

The first report of the controlling immune gene by lncRNAs in innate immune responses was observed by Guttman et al. in 1999 (63, 64).

8) lncRNA and cellular immune response

The heading should be reconsidered. What is the difference to the section before?

Again, a better organization and language editing is needed!

9) Adatative immunity

„Adatative“ or „adaptative” should be “adaptive”.

Punctuation and language editing needed once again, e.g.:

Briefly, lymphocytes that were knocked down showed down-regulation of Ccr1 to Ccr5 (essential chemokine receptor for migration), moreover, the authors showed that CD45+ cells with low lincrR-Ccr2-5AS expression displayed impaired migration to the lungs in mice model (91).

à what was knocked down in the lymphocytes?

à Full stop and continuation with the next sentence: Moreover, the authors showed

The NeST (nettoie Salmonella pas Theiler’s [cleanup Salmonella not Theiler’s] showed in a murine model that binds to WDR5 (histone component H3 lysine 4 methyl-transferase complex promoting the alteration of histone 3 methylation at the IFN-γ locus, thus the authors conclude that this lncRNA regulates epigenetic marking of IFN, promoting susceptibility to a viral and bacterial pathogen (93).

à several issues in this one sentence…

It is important to mention that the immune and adaptative response works together…

à The authors probably want to say that the innate and the adaptive immune responses work together…right?

10) Immune checkpoints and lncRNA

Currently, there are available antibodies against PD-L1 that showed great success in melanoma, leukemia, and lymphoma, however, their efficacy in solid tumors is limited (97).

à I would argue that several clinical studies that are currently conducted in several solid tumor types show very promising results, especially in lung cancer.

The computational modeling and the simultaneous analysis of several cellular parameters allow for obtaining valuable information on immune checkpoints that probably can be used in clinical settings for the treatment of particular cancer patients shortly (101-103).

à A sentence without any meaningful information.

11) Challenges in using lncRNAs for immunotherapy

Moreover, the subsequent activity in the cytoplasm or nucleus is not an easy task during the design of therapeutic strategies for cancer treatment that involve the use of lncRNA.

à This needs a bit more detailed explanation to understand the challenge herein…

However, it is necessary more studies in the context of ncRNA and lentiviral vectors due to the theoretical possibility of insertion of oncogenes, however, this has not been demonstrated until now (110).

à Besides language editing that is needed here, the logic of the argument is not clear. Why is the risk of integration of a CAR gene (mentioned in the sentence before) and a lncRNA different?

However, the composition that allows the correct delivery in the nucleus is a continuous challenge. Nevertheless, the recent application of ASO-gold-TAT NPs targeting lncRNA MALAT1 in lung cancer noticeably suppressed tumor metastasis in an animal model, prolonging the survival of the animal until 80% (112).

à What is meant by “survival of the animal until 80%”?

Besides this example, there are several sentences in this section that need language editing!

12) Figures

The Figures are overloaded and not very intuitive. They need to be improved and additional figures that highlight selected mechanisms of lncRNAs in regulating tumor and immune cell interactions should be created. Moreover, the use of BioRender software needs to be acknowledged based on the software licensing agreement. The authors should check their subscription and make sure to have the right to use the figures.

Comments on the Quality of English Language

A massive re-writing is needed in order to improve grammar, punctuation and readability of the manuscript.

Reviewer 2 Report

Comments and Suggestions for Authors

Significance: 

A review by Martinez-Castillo Moises et.al. entitled “Immune Modulatory Long-non-Coding RNA as a potential therapeutic target in Cancer” is highlighting the advanced use of lncRNAs as candidates for immune modulation in various carcinogenesis which will be relevant to the field.

Comments:

·        Authors didn’t explain the ability of lncRNAs as producers of micro peptides and their importance and relevance as therapeutic targets.

·        LncRNAs function by various mechanisms including working as scaffolds, decoys, ceRNAs, or interaction with various RNA binding proteins authors can highlight this with individual examples.

Reviewer 3 Report

Comments and Suggestions for Authors

In this manuscript, Moises and colleagues describe the role of lncRNAs in the tumor microenvironment and the regulation of tumor immune response. After an overview of tumor immunity and the role of lncRNA in cancer progression, the authors focus more specifically on the role of lncRNAs in innate and adaptive immunity and their potential effects in tumor immunoediting. The structure of this review article is well-developed, and it highlights some of the important aspects of lncRNAs in the tumor immune microenvironment. However, the number of lncRNAs discussed seems limited, and some lncRNAs important in immune regulation, such as NEAT1, LIMIT, and NKILA, should be mentioned. Moreover, since this topic has been published in previous review articles, the authors should focus on the most recent literature. The manuscript would also need some revision for language and grammar, especially in the abstract and introduction.

Specific comments:

·       In the abstract, line 17, the authors define ncRNAs as “small transcript sequences”. This is not correct, since lncRNAs are also ncRNAs.

·       In the introduction, line 39, ribosomal RNA is included among the small ncRNAs. However, by definition, small ncRNA are transcripts shorter than 200 nt. Only 5S and 5.8S ribosomal RNA are less than 200nt, the other rRNAs are long transcripts.

·       In the overview of tumor immunity, line 80, the authors refer to M2 macrophages without explaining what M2 macrophages are.

·       In line 85, when authors talk about environmental factors, they should give some examples of what factors they are referring to.

·       Figure 1 is confusing and not very informative. Authors should try to highlight better the role of lncRNAs. Maybe multiple figures with specific lncRNA functions in different cells or highlight some specific lncRNAs and their function. Adding a table with immune-related lncRNAs would also help.

·       Since the manuscript focuses on immune modulatory lncRNAs, in the paragraph “lncRNAs as tumor biomarker” it would be more appropriate to discuss lncRNAs that are associated with patient response to immunotherapy, immune-related lncRNAs signatures, etc.

·       Paragraph 5 (line 216-233) does not actually talk about lncRNAs that can activate antitumor immunity. If this is just an introduction to the other paragraphs, maybe the subsequent paragraphs should be labeled as subparagraphs of 5.

·       Some of the biological processes should be better explained. For example, in paragraph 9, the authors say: “The cell signaling of tumor cells promotes the downregulation of immunoreceptors, that in consequence causes the over-stimulation of inhibitory immunoreceptors.” In which cells the downregulation of immunoreceptors is happening? How the downregulation of these immunoreceptors can “over-stimulate” inhibitory immunoreceptors?

·       On page 9, line 400, they write: “The authors found that PD-L1 lncRNA splice isoform…”. Who are the authors? Better to use the name of the authors.

·       Line 403, I think PD-Li is a typo. Should it be PD-L1?

·       The paragraph “challenges in using lncRNAs for immunotherapy” would require improvements. For example, I don’t understand why the fact that lncRNAs are important in several biological processes should make it challenging their delivery (lines 424-426). Also, the authors focus mostly on the delivery of ncRNAs. However, since several of the lncRNAs have a pro-tumor function, what about the challenges in targeting these lncRNAs?

Comments on the Quality of English Language

Revision for English language and grammar are needed, especially in the abstract and introduction.

Round 2

Reviewer 1 Report

Comments and Suggestions for Authors

The authors present a significantly improved manuscript in which they largely addressed my previous concerns. Since the manuscript is still a wild mix of diverse and partially unrelated information, I have the following suggestions for further improvement:

a) Please correct the typo in the title (Targts misses an “e”). Please check the whole manuscript for typos again.

b) Carefully go over the abstract again. There is much redundancy in it.

c) Reconsider the granularity and content of your sub-headings/paragraphs. It is way too much and some paragraphs do not contain a meaningful message. Paragraph 2.4. “Cancer Vaccines and Adoptive T-Cell Therapy” is just one example: 4 lines of blah-blah.   

d) Paragraph “2.7 ncRNAs and Immune response” should be the start for a new part/paragraph of the review and should be connected to the subsequent parts describing the role of ncRNAs in cancer immunity.

e) Paragraph 3 “lncRNAs and cancer” with its generic content highlighting general functions of lncRNAs in cancer does not really fit in here and/or should be merged with/shifted to the introduction.

f) What is the purpose of paragraph 4 “lncRNAs as tumor biomarkers”. If you want to highlight their biomarker potential, I would suggest to keep the focus on immune-related biomarker functions.

g) The content in paragraph 5 should be at the center of the review. However, the information contained herein is very limited and hardly any precise examples are given. If there is so little known about the role of lncRNAs during immune responses, one might question the relevance of this review.

Again, consider your sub-headings and try to merge this paragraph with the latter ones.

h) Reconsider your headings and/or the content of your paragraphs 6-8. The innate immune response is mounted by cells as well. Therefore, a sub-heading like “Cell-mediated immunity and lncRNAs” is difficult to include and set apart from paragraph 6 and 8.

i) In the Conclusion and Perspectives paragraph there are two enumerations, both starting with the number 1. Please change the second enumeration and use letters instead of numbers.

Overall, the manuscript contains lots of information that is not really necessary and some parts are even unsubstantial due to the lack of relevant details and examples. Thus, you might want to critically reflect on each and every paragraph and its relevance for the overarching topic of the review. Sometimes it is better to shorten than to add something, i.e. reduce the fraction of blah-blah. Importantly, use sub-headings carefully and make sure the content matches the expectation.  

Comments on the Quality of English Language

Check for typos again.

Author Response

We would like to thank the reviewer for his comments and suggestions to improve our manuscript.

Reviewer 3 Report

Comments and Suggestions for Authors

While the manuscript has improved in English readability and added some new lncRNAs, it still needs major improvements in the content and structure. There are also mistakes in the description of some biological processes. Some of the references are missing. The new version contains too many subparagraphs, some with just two or three sentences, making the review fragmented and unpleasant to read. Some of these subparagraphs contain very general information without providing more detailed examples. Some of the suggested changes in the previous comments have not been made or addressed.

Specific comments:

1.       The abstract repeats the same concept and aim of the review multiple times.

2.       Example of missing references: Sentence lines 84-86 and sentence lines 86-88.

3.       In line 206, the authors say: “They can also create an immunosuppressive microenvironment by secreting factors that inhibit immune responses, such as cytokines or regulatory T cells (Tregs)”. Tumor cells cannot secrete T regs, and T regs are not factors but cells.

4.       Same in line 211: “TME factors such as immune cells”. I would define immune cells as TME components and not “factors”.

5.       Example of subparagraph that needs more explanations: subparagraph 2.4 “Cancer vaccines and adoptive T-cell therapy”. Examples of cancer vaccines? Examples of adoptive T cell therapy?

6.       The lncRNAs associated with immunotherapy response were not discussed.

7.       In paragraph 4.4, it does not seem that the lncRNAs described have been used to monitor treatment efficacy.

8.       What is the connection between the detection of lncRNAs in body fluids and the fact that lncRNAs are overexpressed in tumors? Were these lncRNAs overexpressed in tumors detected in body fluids?

9.       Line 472: LINC00665 did not have an impact on the infiltration rate, but it was associated with the infiltration rate. Also, reading the original paper, it does not seem that LINC00665 functions by targeting FTX.

10.   Line 694: “While experimental use of antibodies against immunosuppressive immune checkpoint inhibitors (ICI) has shown promise…”. This sentence is not correct. Or they say antibodies against immune checkpoints, or they call them immune checkpoint inhibitors (which are the antibodies).

11.   In the “challenges” they have not discussed lncRNA targeting as suggested. Also, it is still not clear why, if lncRNAs promote cancer cell growth I should deliver them.

Comments on the Quality of English Language

English language is acceptable.

Author Response

(The authors gave the same response as above.)
